# Clarithromycin overcomes stromal cell-mediated drug resistance against proteasome inhibitors in myeloma cells via autophagy flux blockage leading to high NOXA expression

**Shota Moriya**[1]*, **Hiromi Kazama**[1], **Hirotsugu Hino**[1,2], **Naoharu Takano**[1], **Masaki Hiramoto**[1], **Shin Aizawa**[2], **Keisuke Miyazawa**[1]*

**1** Department of Biochemistry, Tokyo Medical University, Tokyo, Japan, **2** Division of Anatomical Science, Department of Functional Morphology, Nihon University School of Medicine, Tokyo, Japan

* moriya@tokyo-med.ac.jp (SM); miyazawa@tokyo-med.ac.jp (KM)

**Data Availability Statement:** All relevant data are within the manuscript and its Supporting Information files.

## Abstract

We previously reported that macrolide antibiotics, such as clarithromycin (CAM), blocked autophagy flux, and simultaneous proteasome and autophagy inhibition by bortezomib (BTZ) plus CAM resulted in enhanced apoptosis induction in multiple myeloma (MM) cells via increased endoplasmic reticulum (ER) stress loading. However, in actual therapeutic settings, cell adhesion-mediated drug resistance between bone marrow stromal cells (BMSC) and MM cells has been known to be a barrier to treatment. To investigate whether CAM could enhance BTZ-induced cytotoxicity in MM cells under direct cell adhesion with BMSC, we established a co-culture system of EGFP-labeled MM cells with BMSC. The cytotoxic effect of BTZ on MM cells was diminished by its interaction with BMSC; however, the attenuated cytotoxicity was recovered by the co-administration of CAM, which upregulates ER stress loading and NOXA expression. Knockout of NOXA in MM cells canceled the enhanced cell death by CAM, indicating that NOXA is a key molecule for cell death induction by the co-administration of CAM. Since NOXA is degraded by autophagy as well as proteasomes, blocking autophagy with CAM resulted in the sustained upregulation of NOXA in MM cells co-cultured with BMSC in the presence of BTZ. Our data suggest that BMSC-associated BTZ resistance is mediated by the attenuation of ER stress loading. However, the addition of CAM overcomes BMSC-associated resistance via upregulation of NOXA by concomitantly blocking autophagy-mediated NOXA degradation and transcriptional activation of NOXA by ER stress loading.

## Introduction

Multiple myeloma (MM) is a refractory hematological malignancy characterized by monoclonal gammopathy. Bortezomib (BTZ) is a first-line 26S proteasome inhibitor that was approved by the FDA in 2003 for the treatment of MM [1, 2]. MM cells are constantly exposed to

**Funding:** This study was supported by funds provided through the NEXT-supported program of the Strategic Research Foundation at Private Universities (S1411011, 2014–2018) from the Ministry of Education, Culture, Sports, Science and Technology of Japan to K.M., Grant-in-Aid from Tokyo Medical University Cancer Research to S.M., and JSPS KAKENHI (Grant Number 22K06653) to S.M. All funder had no role in study design, data collection and analysis, the decision to publish or preparation of the manuscript. All funders provided consumable costs for this study.

**Competing interests:** The authors have declared that no competing interests exist.

endoplasmic reticulum (ER) stress, which is the accumulation of misfolded proteins inside the ER lumen due to their potent abnormal protein production. In response to ER stress, a coordinated cellular response known as the unfolded protein response is evoked in normal cells to reduce the number of proteins that enter the ER by suppressing the translational rate and to increase the folding capacity of the ER via translational activation of chaperone proteins. If proteins cannot be folded correctly in the ER, they are translocated to the cytoplasm for degradation via the ubiquitin-proteasome pathway, a process termed ER-associated degradation (ERAD), for adaptation. However, if these adaptive strategies fail, apoptosis is triggered by the induction of the proapoptotic transcription factor CHOP (CCAAT-enhancer-binding protein homologous protein)/GADD153 and IRE1, which are involved in signaling via caspase-12 [3]. Therefore, cellular homeostasis of MM cells is believed to be highly dependent on intracellular protein processing systems, such as the ubiquitin-proteasome system, because of their potent protein production [4]. We and others have reported that proteasome inhibition by BTZ and carfilzomib (CFZ) results in the accumulation of misfolded proteins, leading to the induction of apoptosis in MM cells via ER stress, including CHOP induction [4, 5].

It is well known that the bone marrow microenvironment plays a critical role in MM cell progression as well as drug resistance. Bone marrow stromal cells (BMSC) directly interact with MM cells to produce several adhesion molecules, extracellular matrix (ECM), and soluble factors such as IL-6, IGF-1, and SDF-1, which promote cell adhesion between BMSC and MM cells, as well as MM cell growth. Integrin VLA-4, expressed on the surfaces of MM cells, is known to lead to adhesion to the BMSC of MM cells by interacting with ligands on BMSC, VCAM-1 and fibronectin [6, 7]. Many studies have shown that MM cells become resistant to chemotherapeutic agents by directly interacting with BMSC, thereby leading to poor survival of patients with MM [7–10]. This type of drug resistance is designated as cell adhesion–mediated drug resistance [11, 12]. It has been reported that a series of anti-apoptotic and pro-survival pathways are activated in cell adhesion–mediated drug resistance, including the IGF-1R/PI3K/Akt and MAPK/ERK pathways, as well as epigenetic regulation such as inhibition of histone H3-lysine 27 (H3K27) methylation [13]. However, the network between BMSC and MM cells is extensive, complex, and requires further investigation. Recently, proteasome subunit beta type 5 (PSMB5) mutations were identified as one of the causes for drug resistance in MM cells chronically exposed to BTZ [14]. That means cell adhesion-mediated drug resistance prolongs the period of BTZ treatment and contributes to the acquisition of drug resistance due to the genetic mutations in the MM cells. Therefore, overcoming of stromal cell mediated drug resistance is important to prevent genetic drug resistance caused by chronic chemotherapy.

Macroautophagy (hereafter, autophagy) is a highly conserved eukaryotic cellular process in which cellular contents are degraded by lysosomes and recycled [15]. Misfolded proteins of the ER are ubiquitinated and selectively degraded in the cytosol mainly by the proteasome, a process collectively termed ER-associated degradation. However, under conditions of proteasome inhibition or the accumulation of misfolded proteins beyond proteasome degradation ability, the misfolded proteins can also be degraded by the autophagy-lysosomal pathway [16]. This process is mediated by docking proteins such as p62/SQSTM-1 and NBR1, which contain a ubiquitin (Ub) -related domain and an LC3 (Atg8) interaction region. LC3-II (the lipidated form of LC3-I) is located in the inner and outer membranes of autophagosomes. Thus, after binding to p62, Ub proteins are incorporated into autophagosomes via the p62–LC3-II interaction [17]. We previously reported that macrolide antibiotics, including clarithromycin (CAM), have an inhibitory effect on autophagy flux and simultaneous inhibition of two major intracellular degradation systems, the ubiquitin-proteasome inhibition by BTZ and the autophagy-lysosome inhibition by CAM systems, which remarkably enhances ER stress-mediated apoptosis along with upregulation of CHOP/GADD153 in MM cell lines and breast cancer cell

lines *in vitro* [18, 19]. This suggests a MM therapeutic possibility by simultaneous proteasome and autophagy inhibition. However, it remains unclear whether this ER stress loading can be reproduced in the bone marrow microenvironment. It has also been reported that the combination regimen of CAM (Biaxin), lenalidomide (Revlimid), and dexamethasone (BiRD) is effective with manageable side effects in the treatment of symptomatic, newly diagnosed MM [20], although CAM used as a single agent in advanced myeloma has been reported to be ineffective [21, 22]. Therefore, the therapeutic effect of CAM for patients with MM is still controversial.

In the present study, we attempted to establish a co-culture system of MM and stromal cells to allow direct cell-cell interactions. In our co-culture system, the cytotoxic effect of BTZ/CFZ on MM cells was significantly inhibited in the presence of stromal cells via direct cell-cell interaction. However, the results clearly demonstrate that CAM overcomes this stromal cell-mediated drug resistance in MM cells by NOXA upregulation. This is mediated through transcriptional upregulation by ER stress loading and by blocking the degradation of NOXA by simultaneous inhibition of the ubiquitin-proteasome and autophagy-lysosome systems.

## Materials and methods

### Reagents

Bortezomib (BTZ) and carfilzomib (CFZ) were purchased from Selleck Chemicals (Houston, TX, USA) and dissolved in dimethyl sulfoxide (DMSO; Nacalai Tesque, Kyoto, Japan) for stock solution at 1 mM. Clarithromycin (CAM) was purchased from Tokyo Chemical Industry (Tokyo, Japan) and dissolved in ethanol (Nacalai Tesque) for stock solution at 5 mM. Cycloheximide was purchased from Calbiochem (Darmstadt, Germany). Propidium Iodide (PI), puromycin dihydrochloride, G-418 sulfate, and bafilomycin A1 were obtained from the FUJIFILM Wako Pure Chemical Corporation (Osaka, Japan).

### Cell lines and culture conditions

For this study, two cell lines, IM-9 and RPMI8226, derived from MM patients were obtained from the American Type Culture Collection (ATCC) (Manassas, VA, USA). IM-9 has the lymphoblastic features with secreting IgG. The human MM patient-derived cell line KMS-12-PE was obtained from the Japanese Collection of Research Bioresources (JCRB) (Osaka, Japan). The human BMSC lines LP101 and AA101 were established by transfection with a recombinant SV40-adenovirus vector. LP101 was obtained from healthy volunteers and AA101 was obtained from a patient with aplastic anemia. LP101 seems to be of macrophage origin, and AA101 shows fibroblastic characteristics [23]. Human embryonic kidney 293T cells, which are commonly used in biology for protein expression, were obtained from the ATCC. IM-9, RPMI 8226, and KMS-12-PE cells were cultured in RPMI-1640 medium (#R8758; Sigma-Aldrich, St Louis, MO, USA) supplemented with 10% heat-inactivated fetal bovine serum (FBS) (South America, Lot. 42Q8275K; Gibco, Gaithersburg, MD, USA) and penicillin-streptomycin solution (#168–23191; FUJIFILM Wako). LP101 and AA101 cells were cultured in Iscove's Modified Dulbecco's Medium (IMDM) (#11506–05; Nacalai Tesque) supplemented with 10% FBS and penicillin-streptomycin solution. 293T cells were cultured in Dulbecco's Modified Eagle's medium (DMEM) (#D6429; Sigma–Aldrich, Germany) supplemented with 10% FBS. In the co-culture experiments of MM cells with BMSCs, the cells were cultured in a mixed medium containing equal amounts of RPMI-1640 and IMDM supplemented with 10% FBS and penicillin-streptomycin solution. All cell lines were cultured in a humidified incubator containing 5% $CO_2$ and 95% air at 37°C. All cell line experiments were conducted within 10 passages of

thawing. Mycoplasma contamination was tested routinely using the e-Myco™ Mycoplasma PCR Detection kit ver.2.0 (iNtRON Biotechnology, Inc., Korea).

## Establishment of EGFP stably expressing MM cell lines

The pEGFP-N1 plasmid vector (#6085–1) was purchased from Clonetech (Cambridge, MA, USA). MM cells, RPMI8226 and IM-9, were transfected with pEGFP-N1 using the Super Electroporator NEPA 21 (Nepa Gene Co. Ltd., Chiba, Japan) according to the manufacturer's instructions. Forty-eight hours after transfection, the cells were selected for limited dilution cloning in the presence of 400 μg/ml G418, and EGFP strongly positive cells were selected. The fluorescence intensity was confirmed using a digital microscope BZ-X800 (Keyence Co., Osaka, Japan). After cloning, EGFP-stably expressing MM cell lines (RPMI8228/EGFP clone #11 and IM-9/EGFP clone #3) were established and used for subsequent experiments. These cell lines were maintained in RPMI-1640 medium supplemented with 10% heat-inactivated FBS and penicillin-streptomycin solution, as described in the culture conditions section of the Materials and Methods.

## Assessment of viable number of cells

The number of viable cells was assessed using CellTiter Blue, a cell viability assay kit (#G8081; Promega, Madison, WI), according to the manufacturer's instructions, as previously described in detail [18]. Alternatively, flow cytometry was performed to assess viable/nonviable cells using an Attune Acoustic Focusing Cytometer (Life Technologies). For the co-culture of MM cells with BMSC, BMSC were pre-cultured in 24-well plate for 24 h. After obtaining a confluent feeder layer, the MM cells were seeded on the BMSC layer. After treatment with BZ and/or CAM for 48 h, the cells were collected and resuspended in 4 ml of PBS. EGFP-positive viable MM cells were assessed using an Attune® Acoustic Focusing Flow Cytometer at a flow rate of 100 μl/min for 1 min. Propidium iodide (PI) staining was performed as described previously [18].

## Immunoblotting

Immunoblotting was performed as described previously [18]. Briefly, cells were lysed with RIPA lysis buffer (#08714–04; Nacalai Tesque) supplemented with a protease and phosphatase inhibitor cocktail (#07574–61; Nacalai Tesque). Equal amounts of protein were loaded onto the gels, separated by SDS-PAGE, and transferred onto Immobilon-P membranes (#IPVH00005; Merck Millipore, Billerica, MA, USA). The membranes were probed with primary antibodies (Abs), such as ATF3 monoclonal (m) Ab (#sc-518032; 1:1,000), anti-ATF4 mAb (#sc-390063; 1:1,000), anti-GRP78 Ab (#sc-13968; 1:1,000), anti-NOXA mAb (#sc-56169; 1:1,000), anti-p62 (SQSTM1) mAb (#sc-28359; 1:1,000), and anti-GAPDH mAb (#sc-32233; 1:1,000), which were all purchased from Santa Cruz Biotechnology (Santa Cruz, CA, USA). Anti-Atg5 Ab (#12994; 1:1,000), anti-Mcl-1 Ab (#5453, 1:1,000), anti-CHOP mAb (#2895, 1:1,000), anti-PARP Ab (#9542, 1:1000), anti Phospho-eIF2α (Ser51) Ab (#9721; 1:1,000), anti-cleaved caspase-3 (Asp175) Ab (#9661; 1:1,000) were purchased from Cell Signaling Technologies (Danvers, MA, USA), and anti-LC3 Ab (#NB600-1384; 1:1,000) from Novus Biologicals (Littleton, CO, USA). Anti-FLAG mAb (#F3165; 1:1,000) was purchased from Sigma-Aldrich. Immunoreactive proteins were detected with horseradish peroxidase-conjugated secondary antibodies (anti-mouse: #115-035-003, anti-rabbit: #711-035-152; Jackson ImmunoResearch Laboratories, West Grove, PA, USA) and an enhanced chemiluminescence reagent (#WBKLS0500; Merck Millipore). Protein bands were imaged and analyzed using a

Luminograph III System (ATTO Corporation, Tokyo, Japan). Band intensities were analyzed using the CSAnalyzer4 software version 2.3.1 (ATTO).

## Gene expression analysis

Total RNA was extracted from cell pellets using the NucleoSpin RNA Plus kit (Takara Bio, Inc., Otsu, Japan). cDNA was synthesized using the PrimeScript RT Master Mix (#RR036B; Takara Bio, Inc.) according to the manufacturer's instructions. Quantitative real-time PCR was performed on a CFX Opus 96 Real-Time PCR system (Bio-Rad Laboratories, Hercules, CA, USA) using TB Green Ex Taq II (Tli RNaseH Plus) (#RR820B; Takara Bio, Inc.). The data were analyzed using the CFX Maestro software ver. 2.3 (Bio-Rad). The comparative 2(−ΔΔCq) method was used for the relative quantification of gene expression. The expression ratio was standardized using GAPDH as an internal control. The primer sequences are described in S1 Table.

## Establishment of knockout cells

Target sequences for CRISPR interference of NOXA were designed using CRISPR Direct (http://crispr.dbcls.jp/), provided by the Database Center for Life Science (Chiba, Japan). The target sequences were as follows: human NOXA, CGGCACCGGCGGAGATGCCTGGG, human ATG5, AAGAGTAAGTTATTTGACGT [24]. The non-targeting control sequence was GTAGCGAACGTGTCCGGCGT [24]. Oligonucleotides containing a guide RNA (gRNA) sequence were cloned into the pSpCas9(BB)-2A-Puro (PX459) V2.0 vector (#62988; Addgene, Cambridge, MA, USA) [25]. Cells were transfected with the pX459-gRNA vector using the Super Electroporator NEPA 21 (Nepa Gene Co., Ltd) according to the manufacturer's instructions. Beginning the day after transfection, the cells were treated with puromycin (0.5 μg/ml for EGFP-labeled MM cell lines, 2 μg/ml for 293T cells) for two days. Single-cell clones were isolated by limiting dilution, which was detected by immunoblotting using target-specific antibodies to select gene-depleted clones. After successfully obtaining cells, they were used for subsequent experiments.

## Co-culture experiments using cell culture inserts

To extract proteins from MM cell lines co-cultured with BMSC, we used a culture system devised using a cell-culture insert, as previously reported [13]. In brief, LP101 cells were pre-cultured on the reverse side of the membrane of a high pore cell culture insert (#657640; pore size: 0.4-um; pore density: $1 \times 10^8$ cm²; Greiner, Frickenhausen, Germany) for 24 h. After obtaining a confluent feeder layer, the cell culture insert was inserted into a 6-well plate (#657160; Greiner), and MM cells, such as RPMI8226 and IM-9, were seeded on the opposite side of the layer where LP101 cells were cultured. Under non-adherent conditions, LP101 cells were cultured at the bottom of wells in a 6-well plate (#657160; Greiner). MM cells without LP101 were used as a single culture. After treatment with BTZ and CAM for 24 h, MM cells were collected by trypsinization and proteins were extracted as described above.

## Transfection of FLAG-NOXA into 293T cells

To obtain the human cDNA library, mRNA was extracted from RPMI8226 cells using NucleoSpin RNA Plus kits (Takara Bio) and reverse-transcribed using SuperScript III reverse transcriptase (Thermo Fisher Scientific). Human NOXA cDNA (NM_021127) was amplified from the cDNA library by RT-PCR using the KAPA HiFi HotStart ReadyMix PCR Kit (Kapa Biosystems, Woburn, MA, USA). NOXA cDNA was subcloned into a linearized

p3xFLAG-ATF6 vector (#11975; Addgene) [26] using an In-Fusion HD Cloning Kit (Takara Bio). 293T cells were transfected with p3xFLAG-NOXA using Super Electroporator NEPA 21 (Nepa Gene) according to the manufacturer's instructions. Forty-eight hours after transfection, cells were used for subsequent experiments.

### Statistical analysis

All quantitative data are expressed as mean ± standard deviation (SD). Statistical analysis of the cell viability assay was performed using two-way ANOVA followed by Bonferroni's multiple comparison test. Statistical analysis of real-time PCR was performed by one-way ANOVA using the CFX Maestro software ver 2.3 (Bio-Rad). For all other assays, one-way ANOVA variance followed by Bonferroni's multiple comparison test was used. Data analysis, except for real-time PCR, was performed using SigmaPlot software (version 12.0; Systat Software, San Jose, CA). P<0.05 was considered to indicate statistical significance.

## Results

### CAM overcomes stromal cell-mediated drug resistance

BTZ and CFZ inhibited cell growth in a dose-dependent manner in all MM cell lines, The MM cell lines exhibited far more sensitivity to the proteasome inhibitors BTZ and CFZ than did the BMSC lines (Fig 1A). BTZ and CFZ have been reported to induce cytoprotective autophagy and apoptosis in MM cells [4, 18, 19, 27], whereas macrolides, including CAM, have been shown to block autophagic flux [18, 28]. Thus, blocking autophagy with CAM leads to pronounced cytotoxicity of proteasome inhibitors in MM cell lines with ER stress loading in *in vitro* culture [18]. To investigate whether combined treatment with a proteasome inhibitor and CAM enhances cytotoxicity in MM cells, even in the bone marrow microenvironment, we attempted to establish a co-culture system using EGFP-labeled MM cells and BMSC lines. Transfection with EGFP itself had little effect on drug sensitivity compared to the parental cell lines. For flow cytometry, we set the gating area for viable cells with propidium iodide (PI) -negative staining (S1A and S1B Fig in S1 File). the dead EGFP-MM cells were flame-outed from this variable gating area and lost EGFP fluorescence. Next, the EGFP-MM cell suspensions were mixed with the stromal cell suspensions to prepare a series of dilutions that were subsequently processed for flow cytometry. The number of EGFP positive viable MM cells was accurately assessed in the presence of stromal cells (S1C and S1D Fig in S1 File).

Using this system, we next examined whether the co-culture of MM cells on the stromal cell layer affects their sensitivity to proteasome inhibitors. As shown in Fig 1B and 1C, MM cells directly interacting with stromal cells showed prominent repression of the cytotoxic effect of BTZ and CFZ as compared with MM cells in the absence of stromal cells. This suggests that direct cell adhesion to stromal cells appears to contribute to drug resistance previously described as "cell adhesion medicated-drug resistance" [12]. CAM treatment suppressed autophagic flux (Fig 2A). Notably, the repression of the cytotoxicity of BTZ/CFZ in MM cells by co-culture with stromal cell lines appeared to be canceled in the presence of CAM (Fig 2B).

### Concomitant treatment of BTZ and CAM enhances NOXA upregulation along with MCL-1 cleavage in MM cells

To investigate the molecular mechanism underlying stromal cell-mediated drug resistance, we first examined ER stress loading in MM cells in a monoculture system (Fig 3A). Both the increased GRP78 expression in RPMI8226 and the increased p-eIF2α in the early phase and the following decrease of it in the late phase in response to BTZ with CAM treatments, suggest

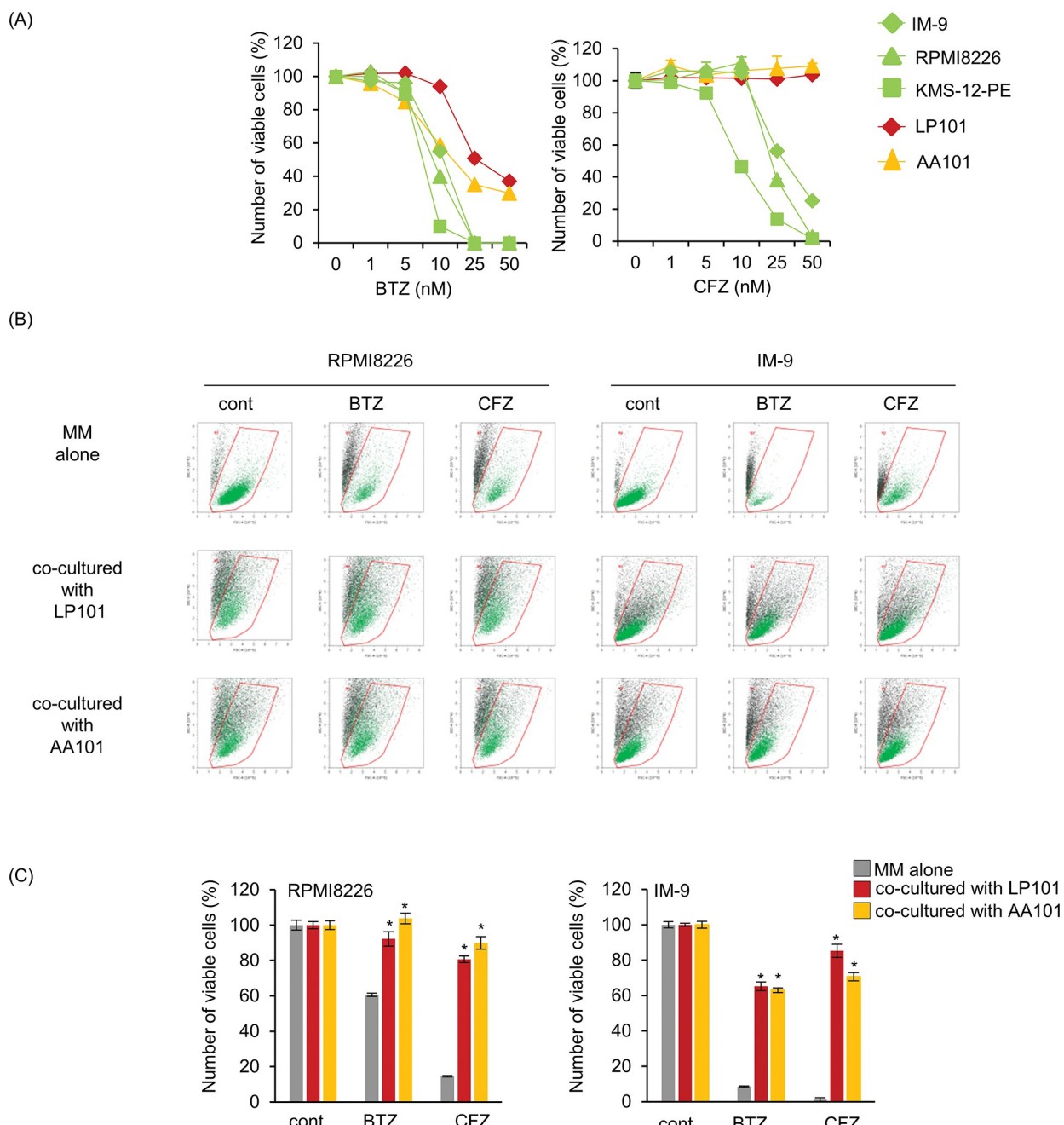

**Fig 1. Stromal cell-mediated bortezomib/carfilzomib resistance in multiple myeloma (MM) cells by co-culture of MM cells with stromal cells.** (A) Cell growth inhibition in response to bortezomib (BTZ)/carfilzomib (CFZ). Human MM cell lines (IM-9, RPMI8226, KMS-12-PE) and human bone marrow stromal cells (BMSC) (AA101, LP101) were treated with BTZ or CFZ at indicated concentrations for 48 h. The viable cell number was assessed using CellTiter Blue. (B) RPMI8226/EGFP and IM-9/EGFP cells were co-cultured with direct adhesion of BMSC (AA101, LP101) in the presence or absence of BTZ (5 nM for RPMI8226/EGFP cells, 10 nM for IM-9/EGFP cells) or CFZ (10 nM for RPMI8226/EGFP cells, 25 nM for IM-9/EGFP cells) for 48 h. The viable cell number was assessed using flow cytometry. The green dots inside of the viable cell gating area (inside of the red line) indicate living MM cells labeled with EGFP. (C) Quantification of Fig 1B. The number of viable cells in each control was described as 100%. Data are presented as the mean ± SD; n = 3; *P<0.05 vs. MM cells without co-culture in stromal cell lines.

enhanced ER stress loading in MM cells. Without stromal cells, simultaneous exposure to BTZ and CAM resulted in enhanced ER stress-related proapoptotic transcription factor CHOP, along with upregulation of the ER stress-inducible transcription factor ATF3 in MM cells.

(A)

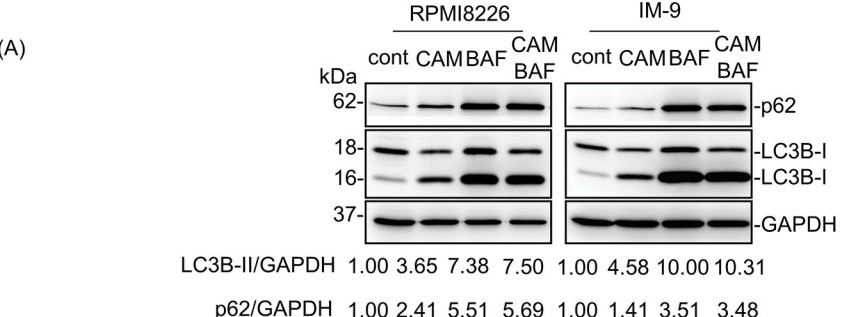

(B)

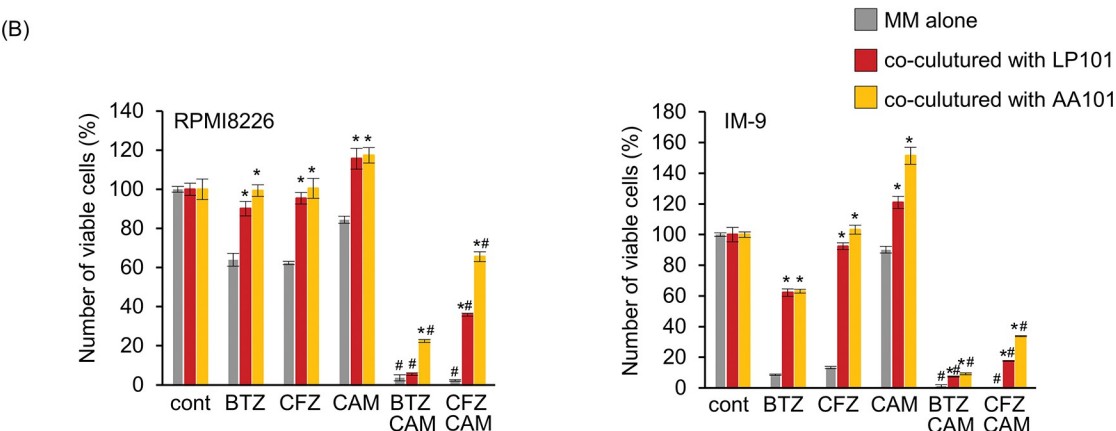

**Fig 2. CAM overcomes stromal cell-mediated drug resistance.** (A) CAM inhibits autophagic flux in MM cells. RPMI8226 and IM-9 cells were treated with CAM (50 μM) in the presence/absence of bafilomycin A1 (BAF, 10 nM) for 24 h. Cellular proteins were immunoblotted with anti-p62 and anti-LC3B Ab. Immunoblotting with anti-GAPDH mAb was performed as an internal control. The numbers indicate the ratio of LC3B-II/GAPDH and p62/GAPDH as a relative ratio to each untreated control. (B) RPMI8226/EGFP and IM-9/EGFP cells were co-cultured with BMSC (LP101, AA101). Cells were treated with BTZ (5 nM for RPMI8226 cells, 10 nM for IM-9 cells) or CFZ (10 nM for RPMI8226, 25 nM for IM-9) in the presence or absence of CAM (50 μM) for 48 h. The viable cell number of MM cells was assessed using flow cytometry. The number of viable cells in each untreated control cell was described as 100%. Data are presented as the mean ± SD; n = 3; *P<0.05 vs. MM cells without co-culture in stromal cell lines.; #P<0.05 vs. each drug alone.

Notably, along with ATF3 upregulation, the expression of proapoptotic BH-3-only protein NOXA was apparently upregulated along with cleavage of the anti-apoptotic protein MCL-1 in both cell lines as compared with those treated with BTZ alone (Fig 3A and S2 Fig in S1 File). Treatment with CAM alone did not result in any changes in ATF3, NOXA, and MCL-1 compared to that of the untreated control cells. Thus, we suggest that the ATF3-NOXA-MCL-1 axis is involved in the enhanced cytotoxicity of the two-drug combination by enhanced ER stress loading. Further, transcriptional activation of GRP78, ATF3, CHOP, and NOXA was observed by the combined treatment of BTZ and CAM, but not in MCL-1, compared with BTZ alone (Fig 3B). The treatment of MM cells with CAM alone did not affect gene regulation or protein expression.

## CAM overcomes stromal cell-mediated drug resistance against proteasome inhibitors by upregulation of NOXA in MM cells

To investigate the molecular mechanisms underlying stromal cell-mediated drug resistance, we used a cell culture insert device, as shown in Fig 4A [13]. This co-culture system showed

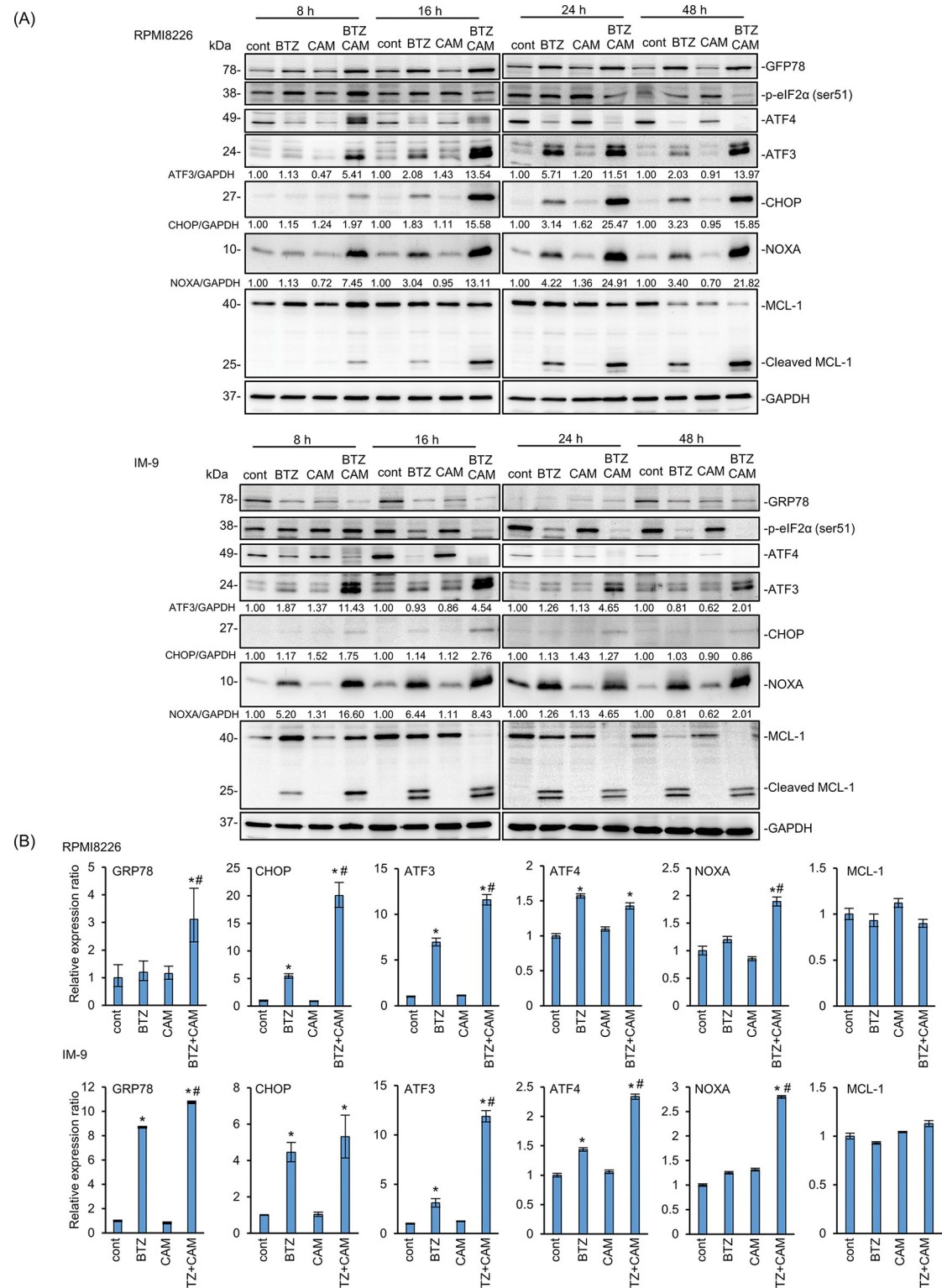

**Fig 3. Enhanced ER stress loading along with NOXA upregulation after simultaneous treatment with BTZ and CAM in MM cell lines.** (A) RPMI8226 and IM-9 cells were treated with BTZ (5 nM for RPMI8226, 10 nM for IM-9) and CAM (50 μM) for the indicated length of time. Immunoblottings using the indicated antibodies were performed to detect unfolded protein response signals. Immunoblotting with anti-GAPDH mAb was performed as an internal control. Numbers indicate the expression level of each protein with GAPDH as an internal control as a relative ratio to each untreated control. (B) RPMI8226 and IM-9 cells were

treated with BTZ (RPMI8226 for 5 nM, 10 nM for IM-9) and CAM (50 μM) for 24 h. The gene expression of GRP78, CHOP, ATF3, ATF4, NOXA, and MCL-1 were assessed by real-time PCR. Data are presented as the mean ± SD; n = 3; *P<0.05 vs. the control; #P<0.05 vs. each drug alone.

that PARP cleavage and caspase-3 activation by BTZ in MM cells were suppressed by adhesion with LP101 cells (Fig 4B). On the other hand, co-culture with stromal cells without adhesion exhibited some repression of cleavage of PARP and caspase-3 in response to BTZ as compared with MM cells without stromal cells. Additionally, BTZ-induced ATF3 and NOXA expression were repressed by LP101 cell adhesion. However, all of the protein repressions in MM cells co-cultured with LP101 cells were canceled by the presence of CAM in the culture medium in both MM cell lines (Fig 4C). This suggests that CAM overcomes stromal cell-mediated drug resistance to BTZ.

It has been reported that NOXA mRNA transcription is regulated by ER-stress inducible transcriptional factor ATF3 [29, 30]. NOXA acts as a pro-apoptotic protein by inhibiting the anti-apoptotic protein MCL-1 in various cancer cell lines [30, 31]. As shown in Figs 3 and 4C, NOXA expression levels were upregulated after co-administration of BTZ with CAM in MM cells, along with ATF3 upregulation at the transcriptional and protein levels. Therefore, we focused on the role of NOXA. We performed CRISPR/Cas9 mediated NOXA KO in IM-9 and RPMI8226 cells (Fig 5A). This resulted in a significant repression of BTZ/CFZ-induced cyto-toxicity (Fig 5B and 5C). Therefore, NOXA appears to be a determinant of BTZ/CFZ sensitiv-ity in MM cells. Notably, at 24-h culture, almost complete cancellation of the enhanced cell death by CAM was observed in NOXA KO BTZ-treated RPMI8226 and IM-9 cells. Further-more, in co-culture with BMSC, NOXA KO MM cell lines all exhibited mitigation of the pro-nounced effect of CAM on BTZ-induced cytotoxicity in NOXA KO MM cells (Fig 5D). These data strongly suggest that CAM overcomes stromal cell-mediated drug resistance by upregu-lating NOXA expression in BMSC-interacting MM cells.

## Upregulation of NOXA in MM cells is mediated through simultaneous inhibition of proteasomal and autophagic degradation

As shown in Fig 3A, CAM further upregulated BTZ-induced NOXA induction. Transcrip-tional activation of NOXA was detected following combined treatment with BTZ and CAM, but not with BTZ or CAM alone (Fig 3B). Recent reports have suggested that NOXA is degraded not only by the ubiquitin-proteasome system but also by the autophagy-lysosomal system [32, 33].

Therefore, in addition to transcriptional activation, it was suggested that the apparent upre-gulation of NOXA in MM cells after BTZ plus CAM treatment appeared to be due to the con-comitant inhibition of proteasomes with BTZ and autophagy with CAM (Fig 3A). To assess NOXA degradation, we performed a chase study; MM cells were exposed to cycloheximide (CHX) to block *de novo* protein synthesis, and NOXA expression was monitored up to 8 h of exposure to BTZ and/or CAM (Fig 6A). Degradation of NOXA was observed within 3 h, even in the presence of either BTZ to block the proteasome or CAM to block autophagy. However, combined treatment with BTZ and CAM resulted in higher NOXA expression in RPMI8226 and IM-9 cells. This strongly suggests that cellular NOXA is concomitantly degraded by autop-hagy and the proteasome. Furthermore, we attempted to establish a FLAG-tagged NOXA expression vector driven by the CMV promoter to exclude other transcriptional regulators and transfected this plasmid into MM cell lines. However, since proapoptotic NOXA overex-pression is lethal for MM cells, many cells died after transfection and could not be used for the experiments. Instead, we used 293T cells derived from human embryonic kidney cells, which

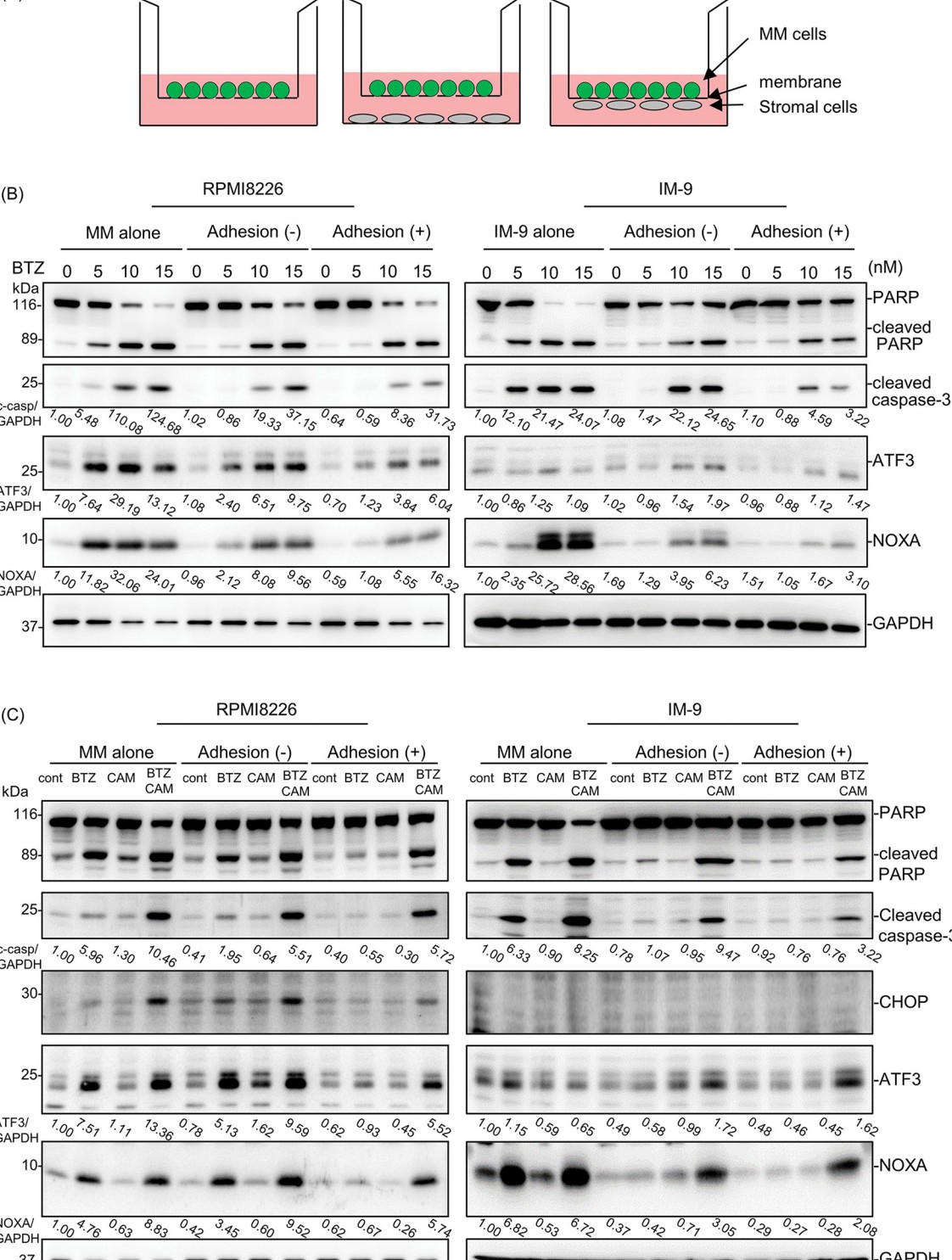

**Fig 4. CAM overcomes stromal cell-mediated suppression of NOXA expression in MM cells induced by BTZ.** (A) Diagram of the co-culture experiment using cell culture inserts. (B) RPMI8226 and IM-9 cells were co-cultured with LP101 using cell culture inserts. Cells were treated with BTZ at indicated concentrations for 24 h. Immunoblotting of cellular proteins from MM cells was performed using the indicated antibodies. Immunoblotting with anti-GAPDH mAb was performed as an internal control. (C) RPMI8226 and IM-9 cells were co-cultured with LP101 using cell culture inserts. Cells were treated with BTZ (5 nM for RPMI8226, 10 nM for IM-9)

and CAM (50 μM) for 24 h. Immunoblotting of cellular proteins from MM cells was performed using the indicated antibodies. Immunoblotting with anti-GAPDH mAb was performed as an internal control. Numbers indicate the expression level of each protein with GAPDH as an internal control as a relative ratio to each untreated control of MM alone.

are widely used for gene transfer experiments. We first established 293T/ATG5 KO cells, which lacked autophagy-inducing abilities (Fig 6B). Then, we transfected FLAG-tagged NOXA plasmids into these cell lines (Fig 6C). ATG5-KO 293T cells exhibited pronounced NOXA and FLAG-NOXA expression. In control 293T cells, the expression of FLAG-NOXA was maintained at a lower level, even though the gene expression of NOXA was enhanced by the CMV promoter. This indicates the rapid degradation of FLAG-NOXA (Fig 6C). Furthermore, in ATG5-KO 293T cells, the addition of 10 nM BTZ alone maintained high FLAG--NOXA expression compared with 293T control cells (Fig 6D). These data indicate that NOXA is degraded by autophagy and proteasomes.

## Discussion

In the present study, we established a co-culture system of MM and BMSC. Using this assay system, we demonstrated that the cytotoxic effect of proteasome inhibitors in MM cells was suppressed in the presence of stromal cells via repression of NOXA and ATF3, ER stress-related transcription factors (Figs 1 and 4A), which appear to represent BMSC-mediated drug resistance [11, 12]. Since KO of NOXA in MM cells resulted in apparent attenuation of the cytotoxicity of BTZ and CFZ (Fig 5), NOXA appears to be a determinant factor of the cytotoxic effect caused by proteasome inhibitors in MM cells. Additionally, we showed that simultaneous treatment with BTZ and CAM extended the half-life of NOXA compared with treatment with either BTZ or CAM alone. This indicated that NOXA degradation depended on both proteasomes and autophagy (Fig 6A). To support this idea, in the presence of BTZ, ATG5 KO cells exhibited a longer half-life of NOXA compared with that in control MM cell lines (Fig 6C). Thus, we concluded that CAM upregulates NOXA by blocking autophagy-mediated degradation as well as stress-mediated transcriptional regulation to overcome BMSC-mediated BTZ/CFZ resistance (Fig 7). These findings are important for clinical application in the management of MM patients by combined treatment with proteasomal and autophagy inhibitors.

It has been reported that concomitant with MCL-1 cleavage and NOXA induction, caspase-3, caspase-8, and caspase-9 are all activated for apoptosis induction by BTZ [34]. Under BTZ treatment, MCL-1/NOXA complexes were highly increased, which led to disruption of MCL-1/Bak complexes and sequential Bax/Bak activation and oligomerization. BAX and BAK oligomers permeabilize the mitochondrial outer membrane to induce cytosolic leakage of cytochrome C from the mitochondria [34]. Additionally, NOXA directly binds to Bax to induce apoptosis [35]. Thus, these data indicate that the NOXA-MCL-1 axis is a critical determinant of apoptosis induction in MM cells in response to proteasome inhibitors. Regarding transcriptional regulation of NOXA, it is well known that p53 regulates NOXA transcription in response to DNA-damaging drugs such as etoposide and doxorubicin [30]. We have previously reported that p53 knockout (KO) A549 cells appear to repress NOXA induction in response to DNA-damaging drugs [36]. Furthermore, the ERAD-mediated ATF3/ATF4 complex binds to the NOXA promoter region to regulate NOXA transcription in mantle cell lymphoma and head and neck squamous cell carcinoma cells [29, 30, 37]. Based on the results of the present study, it is still possible that increased NOXA expression is mediated through the p53 and/or ATF pathways. However, RPMI8226 cells with a p53 missense mutation still exhibited NOXA upregulation (Fig 3). In our study, BTZ treatment increased NOXA protein expression, which was further enhanced by combined treatment with BTZ and CAM. On the

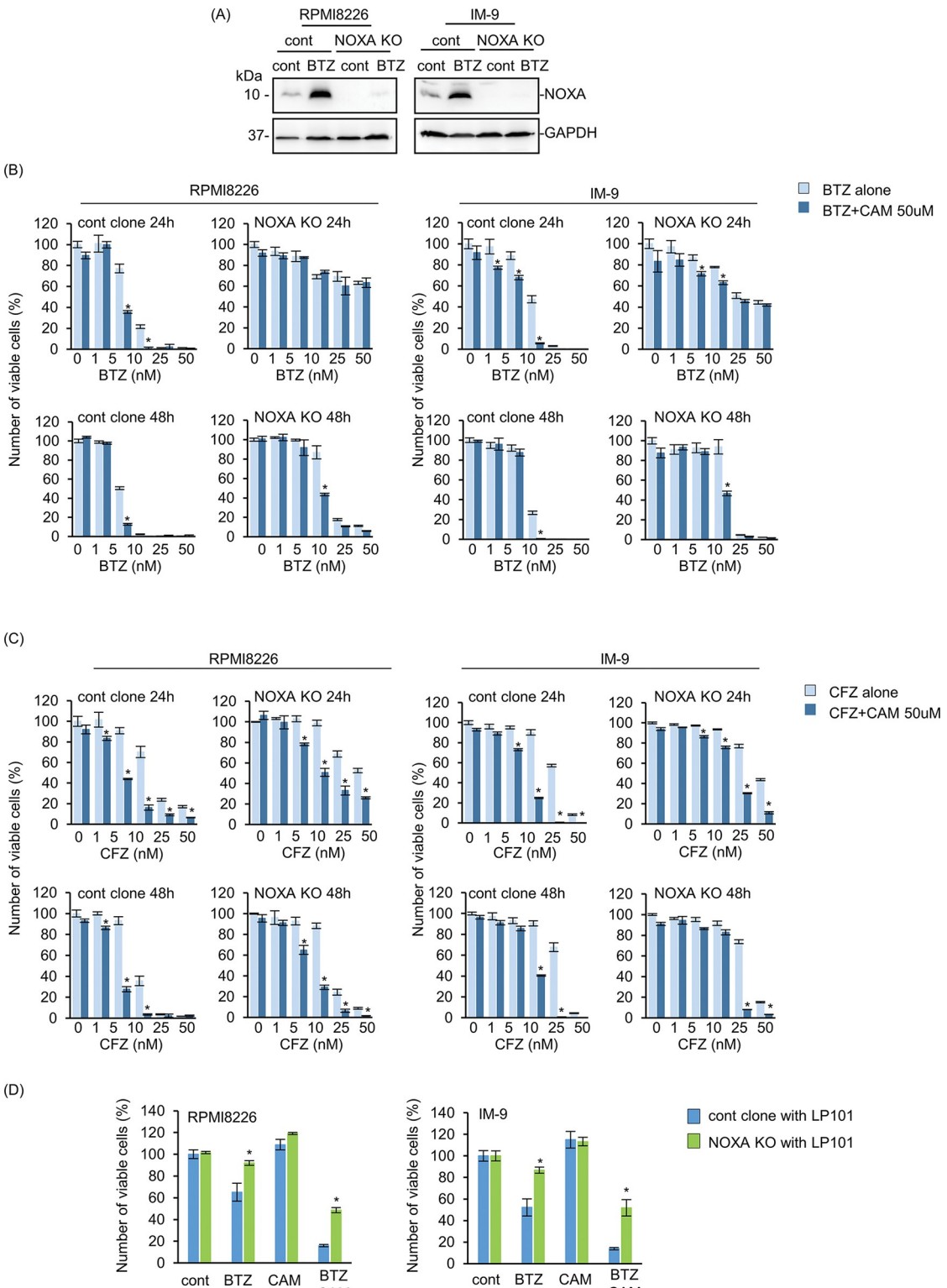

**Fig 5. Involvement of NOXA in the pronounced cytotoxicity of BTZ plus CAM combination in MM cells with/ without stromal cell co-culture conditions.** (A) Confirmation of NOXA-knockout (KO) in RPMI8226/EGFP and IM-9/EGFP cells. Control and NOXA KO MM cells (RPMI8226/EGFP, IM-9/EGFP) were treated with BTZ (10 nM) for 24 h. Cell lysates were immunoblotted with anti-NOXA mAb. Immunoblotting with anti-GAPDH mAb was performed as an internal control. (B, C) Control and NOXA-KO MM cells (RPMI8226/EGFP, IM-9/EGFP) were treated with BTZ (0–50 nM) (B) or CFZ (0–50 nM) (C)

and/or CAM (50 μM) for 24 h and 48 h. The viable cell number was assessed using CellTiter Blue as described in the Materials and Methods. Data are presented as the mean ± SD; *P<0.05 vs. BTZ or CFZ alone. (D) Control and NOXA-KO MM cells (RPMI8226/EGFP, IM-9/EGFP) were co-cultured with LP101. Cells were treated with BTZ (5 nM for RPMI8226 cells, 10 nM for IM-9 cells) and CAM (50 μM) for 48 h. The viable MM cell numbers were assessed using flow cytometry. The number of viable cells in each untreated control cell was described as 100%. Data are presented as the mean ± SD; *P<0.05 vs. control clone.

other hand, NOXA mRNA showed no significant change in response to BTZ, whereas co-administration of BTZ and CAM increased the NOXA mRNA (Fig 3B). We believe that NOXA mRNA was only enhanced in cells treated with both drugs at 24 h because ATF3 was upregulated earlier in BTZ plus CAM-treated MM cells than in BTZ-treated cells.

The expression level of NOXA protein increased as early as 8 h after BTZ plus CAM treatment; therefore, we focused on the degradation pathway rather than the transcriptional activation of NOXA. Indeed, in MM, mantle cell lymphoma, and other solid tumors, including melanoma, a rapid increase in NOXA expression in response to a proteasome inhibitor preceded NOXA mRNA transcriptional activation, indicating its tight regulation by proteasomal degradation [30]. Recently, Cullin-RING E3 ubiquitin ligases were shown to ubiquitinate NOXA lysine 11 residues, leading to proteasomal degradation [30]. In a mouse model, NOXA has an FNLV sequence, which is known as the LC3-interacting region (LIR) motif, suggesting that NOXA is engulfed into autophagosomes via LC3 interaction and subsequently degraded by autophagy [38]. However, there is no FNLV sequence in human NOXA. Wang et al. reported that NOXA is ubiquitinated in K35/K41/K48 and introduced to p62-mediated autophagosome incorporation for autophagy degradation in human non-small cell lung cancer (NSCLC) and colorectal cancer cell lines [32]. Heine et al. reported that the combined treatment of BTZ plus autophagy inhibitor enhances BTZ-induced cell death via blocking autophagic degradation of NOXA in mantle cell lymphoma [33]. In addition, we reported that in lung cancer cell lines, NOXA was upregulated via p53 after treatment with DNA-damaging drugs such as etoposide and doxorubicin, and NOXA expression was further enhanced in the presence of azithromycin, which has a potent autophagy inhibitory effect [36]. Taken together, these data suggest that concomitant transcriptional activation via ATF3/ATF4 by ER stress loading and blocking degradation with BTZ for proteasomes, as well as blocking autophagy with CAM, could induce the maximal NOXA expression state for apoptosis induction in MM cells. In addition, NOXA knockout significantly inhibited the cytotoxic effect of BTZ alone or the combination of BTZ and CAM 24 h after drug administration, but the inhibition was attenuated 48 h after drug administration (Fig 5B). The results indicate that NOXA accumulation leads to cell death up to 24 h after drug administration and also suggests that another cell death pathway mediated by the ER stress-induced transcription factors ATF3 or CHOP may be induced approximately 48 h after drug administration.

As CAM has an inhibitory effect on autophagy (Fig 2A) [18, 28], it is reasonable to assume that autophagy was inhibited in BMSC and MM cells by CAM treatment. Recent reports suggest that microenvironmental autophagy promotes tumor growth. Frassanito et al. reported that bone marrow fibroblasts isolated from MM patients enhanced the secretion of TGF-β and other cytokines to support MM growth by BTZ-induced autophagy and oxidative stress [39]. In pancreatic cancer, interstitial cells produce growth factors via autophagy. Endo et al. reported that pancreatic stellate cells with high autophagic activity, which produce ECM molecules and IL-6, are associated with shorter survival times and disease recurrence in patients with pancreatic cancer [40]. Thus, it was suggested that the autophagy inhibition in pancreatic stellate cells might reduce pancreatic tumor invasiveness by altering the tumor stroma. In the *Drosophila melanogaster* malignant tumor model, transformed cells engage surrounding normal cells as active and essential microenvironmental contributors to early tumor growth

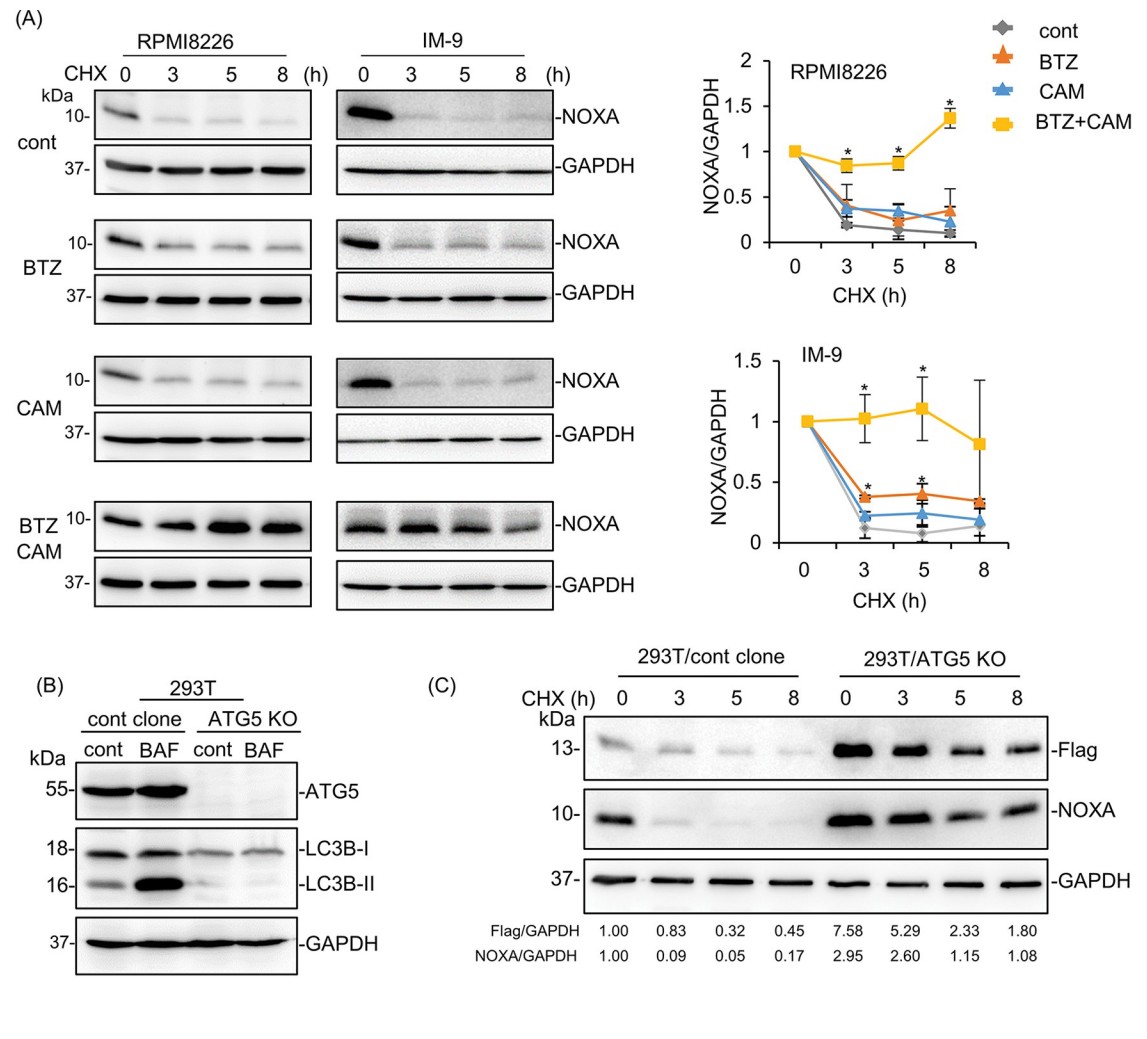

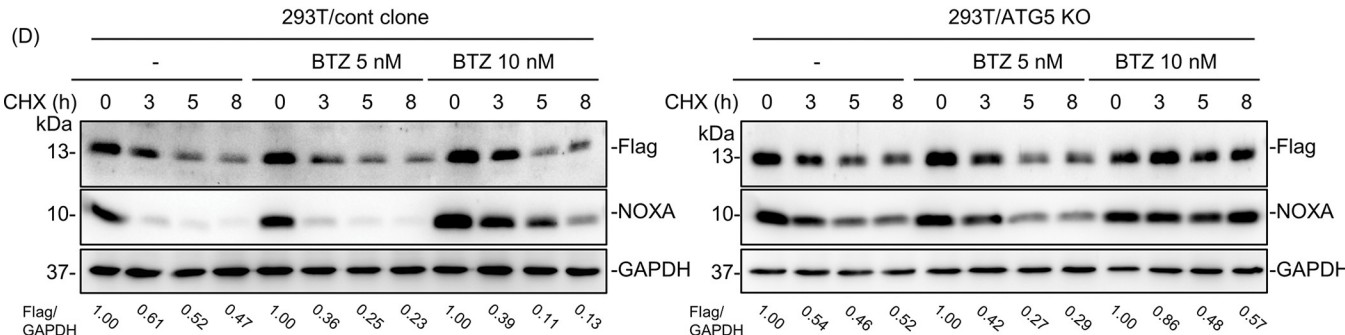

**Fig 6. NOXA degradation via autophagy in MM cells.** (A) RPMI8226 and IM-9 cells were pre-treated with BTZ (5 nM for RPMI8226 cells, 10 nM for IM-9 cells) and/or CAM (50 μM) for 16 h. After pre-treatment, cells were subsequently exposed to cycloheximide (CHX; 50 μM) for the indicated length of time. Cellular proteins were immunoblotted with anti-NOXA mAb. Immunoblotting with anti-GAPDH mAb was performed as an internal control. Band intensities of NOXA/GAPDH were summarized in the right panel. Numbers indicate the relative band intensities of NOXA to GAPDH as compared to 0 h. Data are presented as the mean ± SD; n = 3; *P<0.05 vs. untreated control cells. (B) Control 293T and ATG5-KO 293T cells were treated with bafilomycin A1 (BAF) (10 nM) for 24 h. Cell lysates were immunoblotted with anti-ATG5 and anti-LC3B Ab. Immunoblotting with anti-GAPDH mAb was performed as an internal control. (C) CMV promoter-regulated FLAG-NOXA vectors were transfected into 293T/cont and 293T/ATG5-KO cells. After 48 h-transfection, cells were subsequently exposed to CHX (50 μM) for the indicated length of time. Immunoblotting using FLAG mAb and NOXA mAb was performed. Anti-FLAG Ab indicates FLAG-NOXA and anti-NOXA Ab indicates endogenous NOXA, respectively. Immunoblotting with anti-GAPDH mAb was performed as an internal control. Numbers indicate the relative band intensity of FLAG-NOXA to GAPDH and endogenous NOXA to GAPDH. (D) CMV promoter-regulated FLAG-NOXA vectors were transfected into 293T/cont and 293T/ATG5-KO cells. After 48 h-transfection, cells were pre-treated with BTZ (5 nM, 10 nM) for 16

h. After pre-treatment, cells were subsequently exposed to CHX (50 μM) for the indicated length of time. Immunoblotting using FLAG mAb and NOXA mAb was performed. Anti-FLAG Ab indicates FLAG-NOXA and anti-NOXA Ab indicates endogenous NOXA, respectively. Immunoblotting with anti-GAPDH mAb was performed as an internal control. Numbers indicate the relative band intensity of FLAG-NOXA to GAPDH as compared to 0 h.

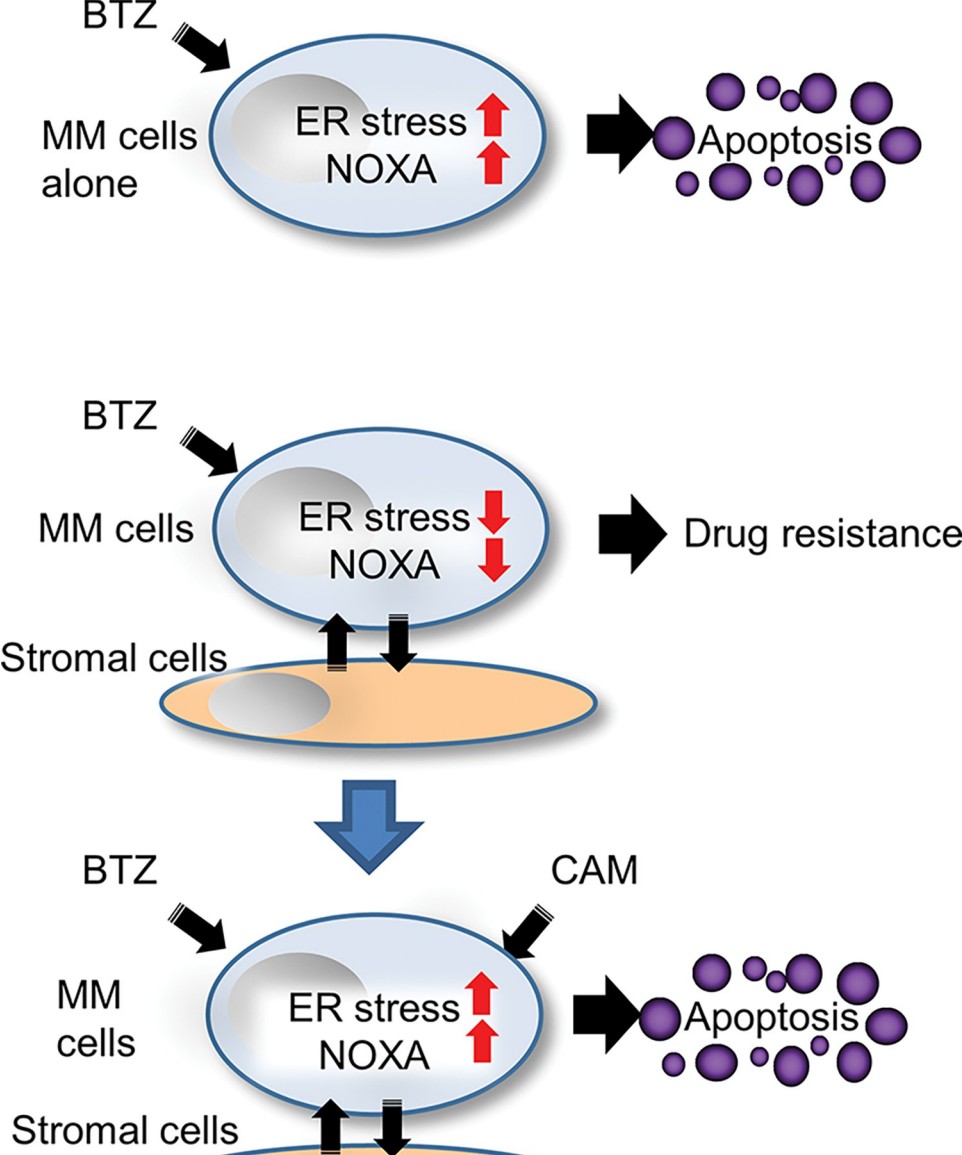

**Fig 7. Molecular mechanism for overcoming stromal cell-mediated BTZ resistance by CAM in MM cells.** In the absence of bone marrow stromal cells, BTZ induces apoptosis in MM cells via ER stress loading, along with the induction of the proapoptotic protein NOXA. In the presence of stromal cells, BTZ-induced cytotoxicity in MM cells was inhibited via cell-to-cell interaction leading to repression of ER stress loading as well as repression of NOXA induction. However, the addition of CAM, an autophagy inhibitor overcomes stromal cell-mediated BTZ resistance by recuperating ER stress loading and by maintaining high expression levels of NOXA by blocking autophagy-mediated NOXA degradation. Notably, NOXA expression is determined by the balance between ER stress-mediated transcriptional upregulation and proteasomal and autophagy-mediated degradation.

through nutrient-generating autophagy. Autophagy in the microenvironment is induced by Drosophila tumor necrosis factor and IL-6-like signaling from metabolically stressed tumor cells, and tumor growth is dependent on the active transport of amino acids produced by autophagy in microenvironmental cells. Therefore, tumor growth can be pharmacologically inhibited by autophagy inhibitors [41]. It was also reported that hepatoma cells secrete growth differentiation factor 15 (GDF15) by enhancing the autophagy of interstitial hepatic stellate cells and promoting the proliferation of hepatoma cells [42]. According to these previous reports, the effects of CAM should also be discussed from the perspective of the microenvironment. Additionally, the mechanism underlying the inhibitory effect of CAM on autophagy is not fully understood. Recently, Petroni et al. reported that CAM inhibits autophagic flux in colorectal cancer by regulating the hERG1 potassium channel interaction with PI3K [43]. CAM has also been reported to inhibit myeloma growth factors such as IL-6 [44]. Therefore, we must also consider the multiple pharmacological mechanisms of CAM that might be involved in overcoming BMSC-mediated drug resistance to proteasome inhibitors. Further study is required, and our experimental system appears to be suitable for this type of study.

Venetoclax is a potent BCL-2 selective BH3-mimetic that is clinically approved for use in chronic lymphocytic leukemia and treatment of patients with acute myeloid leukemia who are aged or unmet for intensive induction chemotherapy [45]. In patients with MM, phase I trials exhibited promising outcomes and led to a placebo-controlled phase III trial of venetoclax combined with BTZ and dexamethasone [46]. In the BELLINI trial, the venetoclax group was superior to the placebo group in terms of progression-free survival, but failed to prolong overall survival due to a higher incidence of treatment-related deaths [47]. Of note, it was reported that the clinical efficacy of venetoclax could be predicted by the ratio of BCL-2/MCL-1 mRNA expression in MM cells [48, 49]. This suggests that concomitant inhibition of BCL-2 by venetoclax and MCL-1 by other drugs induces potent clinical efficacy. Since CAM has been used for a long time as a safe antibiotic in clinical settings, information regarding its pharmacokinetics and adverse events has been accumulated. Our data indicate that by combining the use of BTZ, CAM can potently upregulate NOXA expression, leading to subsequent MCL-1 inhibition. Therefore, a combination of CAM and BTZ, as well as CAM for the venetoclax-based regimen, appears to be another possibility for MM treatment.

## Supporting information

**S1 File. Supplementary information (S1 Fig to S2 Fig and S1 Movie).**
(DOCX)

**S1 Fig. Cytogram image of Fig 2B.**
(TIF)

**S2 Fig. Cytogram image of Fig 5D.**
(TIF)

**S1 Raw images. Raw blot images.**
(PDF)

**S1 Movie. Supplementary movies.** Confocal microscopy for monitoring the interaction between MM cells and stromal cells.
(AVI)

**S1 Table. Primer sequences for real-time PCR.**
(DOCX)

## Author Contributions

**Conceptualization:** Shota Moriya, Keisuke Miyazawa.

**Data curation:** Shota Moriya, Keisuke Miyazawa.

**Formal analysis:** Shota Moriya, Hiromi Kazama, Keisuke Miyazawa.

**Funding acquisition:** Shota Moriya, Keisuke Miyazawa.

**Investigation:** Shota Moriya, Hiromi Kazama, Hirotsugu Hino, Naoharu Takano, Masaki Hiramoto, Keisuke Miyazawa.

**Methodology:** Shota Moriya, Keisuke Miyazawa.

**Project administration:** Shota Moriya, Keisuke Miyazawa.

**Resources:** Shin Aizawa.

**Supervision:** Shota Moriya, Keisuke Miyazawa.

**Validation:** Shota Moriya, Keisuke Miyazawa.

**Visualization:** Shota Moriya, Keisuke Miyazawa.

**Writing – original draft:** Shota Moriya, Naoharu Takano, Masaki Hiramoto, Keisuke Miyazawa.

**Writing – review & editing:** Shota Moriya, Naoharu Takano, Masaki Hiramoto, Keisuke Miyazawa.

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
