## [Decision Letter · Decision Letter 0]

16 Aug 2023

PONE-D-23-14889Clarithromycin overcomes stromal cell-mediated drug resistance against proteasome inhibitors in myeloma cells via autophagy flux blockage leading to high NOXA expressionPLOS ONE

Dear Dr. Moriya,

Thank you for submitting your manuscript to PLOS ONE. After careful consideration, we feel that it has merit but does not fully meet PLOS ONE’s publication criteria as it currently stands. Therefore, we invite you to submit a revised version of the manuscript that addresses the points raised during the review process.

We look forward to receiving your revised manuscript.

Kind regards,

Vladimir Trajkovic

Academic Editor

PLOS ONE

Journal Requirements:

"This study was supported by funds provided through the NEXT-supported program of the Strategic Research Foundation at Private Universities (S1411011, 2014–2018) from the Ministry of Education, Culture, Sports, Science and Technology of Japan to K.M., Grant-in-Aid from Tokyo Medical University Cancer Research to S.M., and JSPS KAKENHI (Grant Number 22K06653) to S.M. "

"This study was supported by funds provided through the NEXT-supported program of the Strategic Research Foundation at Private Universities (S1411011, 2014–2018) from the Ministry of Education, Culture, Sports, Science and Technology of Japan to K.M., Grant-in-Aid from Tokyo Medical University Cancer Research to S.M., and JSPS KAKENHI (Grant Number 22K06653) to S.M. "

"This study was supported by funds provided through the NEXT-supported program of the Strategic Research Foundation at Private Universities (S1411011, 2014–2018) from the Ministry of Education, Culture, Sports, Science and Technology of Japan to K.M., Grant-in-Aid from Tokyo Medical University Cancer Research to S.M., and JSPS KAKENHI (Grant Number 22K06653) to S.M. "       

Reviewers' comments:

Reviewer's Responses to Questions

**Comments to the Author**

1. Is the manuscript technically sound, and do the data support the conclusions?

Reviewer #1: Yes

Reviewer #2: Partly

2. Has the statistical analysis been performed appropriately and rigorously? 

Reviewer #1: Yes

Reviewer #2: N/A

3. Have the authors made all data underlying the findings in their manuscript fully available?

Reviewer #1: Yes

Reviewer #2: Yes

4. Is the manuscript presented in an intelligible fashion and written in standard English?

Reviewer #1: Yes

Reviewer #2: Yes

5. Review Comments to the Author

Reviewer #1: BACKGROUND:

Clarithromycin (CAM), a semisynthetic macrolide antibiotic, is widely used antibacterial drug. Recently, the efficacy of CAM as an add-on drug for treating multiple myeloma (MM) has been noted. It is known that a single treatment of CAM has no efficacy for treating MM. Bortezomib (BTZ), the 26S proteasome inhibitor, has been widely used for treating MM. Blocking of the ubiquitin proteasome system by BTZ leads to the accumulation of unfolded or misfolded protein in the endoplasmic reticulum (ER) in MM cells; this results in ER stress followed by unfolded protein response. The high efficacy of the chemotherapeutic regimen combining CAM with lenalidomide and dexamethasone (BiRD regimen) in treating MM has been reported. Recent studies have demonstrated that CAM is a potent inhibitor of autophagy. Thus, it has been used as a potent adjuvant for refractory/relapsed MM (RRMM) treatment modalities. Recently, the mechanisms of action whereby CAM inhibits MM cell proliferation have been explained as follows: autophagy inhibition, suppression of IL-6 and steroid sparing/enhancing effect.

REVIEWER‘S COMMENTS:

Generally, the refractoriness against treating MM will occur in actual therapeutic settings. In respect of the refractoriness, the authors emphasized that adhesion-mediated drug resistance between bone marrow stromal cells (BMSCs) and MM cells. To prove this possibility, they established a co-culture system of MM cells combined with BMSC, and demonstrated that the cytotoxic effects of BTZ was diminished under the co-culture system. Furthermore, this attenuated cytotoxicity was recovered by co-administration of CAM. For the explanation of this phenomenon, they focus on NOXA expression. NOXA is known to represent a key molecule for cell death. This was proved by NOXA knockout MM cell experiment in this study. Since NOXA is degraded by autophagy as well as proteasomes, the autophagy block by CAM will cause the sustained upregulation of NOXA in MM cells co-cultured with BTZ. It is noteworthy that authors focused on NOXA to explain the mechanism of enhanced cytotoxicity by BTZ co-added with CAM. The therapeutic modality of BTZ combined with CAM in MM treatment could be a promising mode of treatment.

In clinical field, a study demonstrating the effectiveness of BTZ, CAM and lenalidomide, though a case report, was published in 2018 (A novel combination of bortezomib, lenalidomide, and clarithromycin produced stringent complete response in refractory multiple myeloma complicated with diabetes mellitus—clinical significance and possible mechanisms: a case report) J Med Case Rep: DOI 10.1186/s13256-017-1550-6

GENERAL COMMENTS:

My overall impression is that this manuscript provides very important information for researchers and medical doctors treating MM. It should be appreciated in respect of developing the study of MM and the treatments for RRMM.

REQUESTED REVISIONS:

The text seems to be somewhat complicated and difficult to understand. The manuscript could be more precise and compacted. The use of supplemental figures could be avoided. Plain figures, if possible, could appear successively in the text without supplemental ones. Numbering of references could be in conformity to the submission instruction.

Reviewer #2: The article entitled: “Clarithromycin overcomes stromal cell-mediated drug resistance against proteasome inhibitors in myeloma cells via autophagy flux blockage leading to high NOXA expression” by Moriya et al., studies the possible mechanism of this antibiotic in the improvement of anti-MM therapy. In particular it evaluates if this drug interferes with pro-survival signals provided by stromal cells. However, the data provided in this work are not novel enough and the hypothesis and experimental design show significant pitfalls to allow the article be published.

Major objections:

1) The IM-9 cell line used in this study is a lymphoblastoid cell line, not a MM cell line (Pellat-Deceunynk el al., Blood. 86(10): 4001, 1995)

2) To my knowledge, Noxa is not directly degraded by autophagy, but only by the proteasome (Craxton et al., Cell Death and Differentiation 19: 1424–1434, 2012). So, the hypothesis provided by the authors is not reliable.

3) Most of results of the present article were already reported by authors in previous works (IJO 46: 474-486, 2015 and IJO 42: 1541‑1550, 2013)

6. PLOS authors have the option to publish the peer review history of their article (what does this mean?). If published, this will include your full peer review and any attached files.

Reviewer #1: No

Reviewer #2: No

---

## [Author Response · Author response to Decision Letter 0]

26 Sep 2023

To Reviewer #1:

Thank you for your careful reading and helpful comment. We are very pleased the reviewer commented the significance of this study. 

We revised our manuscript as much as possible. Thank you for giving us the opportunity to strengthen our manuscript with your valuable comments and queries.

To Reviewer #2:

Thank you for your careful reading and helpful comment. Your comments were extremely helpful for us. We revised our manuscript as much as possible.

Thank you for giving us the opportunity to strengthen our manuscript with your valuable comments and queries.

---

## [Editor Report · Decision Letter 1]

20 Nov 2023

Clarithromycin overcomes stromal cell-mediated drug resistance against proteasome inhibitors in myeloma cells via autophagy flux blockage leading to high NOXA expression

PONE-D-23-14889R1

Dear Dr. Moriya,

We’re pleased to inform you that your manuscript has been judged scientifically suitable for publication and will be formally accepted for publication once it meets all outstanding technical requirements.

Kind regards,

Vladimir Trajkovic

Academic Editor

PLOS ONE
---

## [Editor Report · Acceptance letter]

24 Nov 2023

PONE-D-23-14889R1 

Clarithromycin overcomes stromal cell-mediated drug resistance against proteasome inhibitors in myeloma cells via autophagy flux blockage leading to high NOXA expression 

Dear Dr. Moriya:

I'm pleased to inform you that your manuscript has been deemed suitable for publication in PLOS ONE. Congratulations! Your manuscript is now with our production department. 

Kind regards, 

on behalf of

Prof. Vladimir Trajkovic 

Academic Editor

PLOS ONE